# Social Networks’ Engagement During the COVID-19 Pandemic in Spain: Health Media vs. Healthcare Professionals

**DOI:** 10.3390/ijerph17145261

**Published:** 2020-07-21

**Authors:** Ana Pérez-Escoda, Carlos Jiménez-Narros, Marta Perlado-Lamo-de-Espinosa, Luis Miguel Pedrero-Esteban

**Affiliations:** Department of Communication, University of Nebrija, 28015 Madrid, Spain; cjimenez@nebrija.es (C.J.-N.); mperlado@nebrija.es (M.P.-L.-d.-E.); lpedrero@nebrija.es (L.M.P.-E.)

**Keywords:** social networks, health media, communication, COVID-19, engagement, YouTube, Twitter, Instagram, health professionals

## Abstract

An increased use of social networks is one of the most far-reaching consequences of the COVID-19 pandemic. Aside from the traditional media, as the main drivers of social communication in crisis situations, individual profiles have emerged supported by social networks, which have had a similar impact to the more specialized communication media. This is the hypothesis of the research presented, which is focused on health communication and based on a virtual ethnography methodology with the use of social metrics. The aim is to understand the relationship established between the population in general and digital media in particular through the measurement of engagement. In this regard, a comparative study was carried out that describes this phenomenon over a period of six months on three social networks: YouTube, Twitter and Instagram, with a sample composed of specialized health media versus healthcare professionals. The results point to a new communications model that opens up a new space for agents whose content has a degree of engagement comparable to and even exceeding that of digital media specialized in health communication. The conclusions show that the crisis of the pandemic has accelerated the transformation of the communication sector, creating new challenges for the communication industry, media professionals, and higher education institutions related to market demands.

## 1. Introduction

The nature of social interactions in the 21st century, transformed by the change into a digital, connected, and globalized context, and concentrated in digital media, has become a phenomenon which has transformed the organization and representation of knowledge [1,2]. This transformation has been so considerable that in the past few years the impact of social networks has resulted in a media metamorphosis, triggering a change not only in the communication-information ecosystem, but also in the roles of the consumers and producers of news, information, and knowledge [3,4]. The worldwide explosion of the COVID-19 pandemic, and the exposure of the entire population to a disease without a specific pharmacological treatment and with exponential levels of infection, have underlined the value of digital media as the preferred communication channel about health [5]. Thus, aside from the traditional media, as the drivers of the social communication thread in situations of crisis, to which they greatly contribute by providing a truthful, informative story [6], other social agents have appeared. These new agents are supported by social networks, and are massively and efficiently contributing to the re-direction of the management of communication in a worldwide pandemic, along with the social instability it has brought [7].

The traditional pattern of the communication-information narrative is enriched with other narratives and channels that take the floor, the social networks. These are sponsored by digital environments that are able to reach vast amounts of the population, and when championed by specific individual profiles, capture a flow of communication that runs parallel to the formal discourse of the media [8]. This phenomenon breaks the traditional study or classification of the information sources in emergency communication situations [9], adding new mediators to the communication scenario. These, when legitimized by the public in general and the digital context in particular, forcefully rise up due to the trust and impact they create. Thus, despite the counterproductive ability of digital media to create hoaxes and fake news, especially during times of crisis [10], an overwhelming mass of followers is imposed on the networks, which promotes an information–communication flow related with communication about COVID-19 and health.

The participation of citizens in digital media as equals, and the creation of an interconnected global dialogue, have provided a growing importance to these digital Agoras. It has shifted the formal communicative discourse, as shown in recent studies conducted in different spheres such as politics, where the participation of the young is strengthened thanks to their participation in social networks [11], as well as the areas of education [12], social life, and leisure [13,14], but also in the area of health [15]. The study presented aims to analyze and describe how citizens can actually have an impact that is equal to traditional media in social networks in the context of the recent COVID-19 pandemic. This work strives to draw a new reality in which social communication is not driven by traditional media alone and where social dialogue is being established by new social agents that have emerged and are encouraged by the power of digital media.

## 2. The Social Networks and Their Reach: The Fifth Power

Without a doubt, the analysis of the current reality of communication is concerned with the understanding of any phenomenon as a function of the impact it has on social networks. If the most common action in the 20th century for finding information was to sit in front of the TV, buy the daily newspapers, or listen to radio information bulletins, in the 21st century, it is more common to type search terms into an internet browser on any device–smartphone, tablet, computer [16]. Social networks have become platforms of constant media flow that connect almost half of the world’s population, thereby becoming more solidified and in direct competition with media and journalists as actors in the information–communication process [17,18].

Their impact and proliferation worldwide have turned ‘homo videns’ into ‘homo socialis’ [19], and the magnitude of these events well deserves the attention of current communication processes. The phenomenon involves half of the world’s population. According to the We are Social report [20] a total of 3.8 billion people are connected to social networks (Figure 1), with an annual average growth of 13% from 2017, which implies an average of 363 million new social network users each year. If the reports from January 2020 showed an increase of 7% in the use of the internet with respect to the previous year in global terms, meaning 298 million new users, the growth in active users of social networks experienced a growth of 9.2% as compared to the previous year, which implies 321 million more users interacting in the networks.

Thus, if the 20th century was marked by great communication empires and news agencies that dominated the world of information—Murdoch, Reuters, Havas, Associated Press, EFE [21]—in the 21st century, the mastery of data falls to the main technology companies. The health crisis has brought to light the hegemony of those who have mastered the flow of information through the main social networks, subduing a communications industry that is overcome by a saturation of information, forecasting a change in the communication business model [22]. As behemoths of big data and gatekeepers facing an avalanche of fake news that began to circulate on the internet due to COVID-19, the great technology companies reacted (Figure 2). Aware of their power to reach and their impact, the seven most-significant companies in the sector—Facebook, Google, Twitter, YouTube, Microsoft, LinkedIn, and Reddit—published a joint declaration on 17 March 2020, in favor of a mutual effort to fight fraudulent and false information, and to support the content of governmental platforms and authorities related to health in the entire world [23]. Of these seven giants, it is interesting to highlight that Google owns, among its multiple services, the YouTube platform, and Microsoft bought LinkedIn in 2016, so the degree of concentration is actually reduced to only five large companies.

The exponential development of Web 2.0, transformed to 3.0 since the arrival of artificial intelligence and big data, through the use of algorithms that personalize the media menu of the users, significantly contributes not only to radically change the consumption of information in all areas and all spheres of knowledge, and specifically of information related to health, but also amplifies the ability to summon and the possibilities of interaction [24]. Its power to capture audiences clearly de-thrones traditional media in the battle for audiences or the selling of the printed word. Although it is true that the ease with which information is transmitted in these communication spaces turns them into the focus of fake news and hoaxes [25,26], it is not less true that their ability to reach audiences has resulted in the proliferation of professional, specialized channels whose work as providers of information has been beneficial and real. Thanks to social networks, professional workers from many fields have had the opportunity to provide the population with vitally-important information, before any other institutional organs or communications media, and in a more direct manner. Here lies the importance of the present research, which analyses some of the main profiles of health communication during the COVID-19 pandemic, as far-reaching social phenomena and legitimate alternatives for the flow of information–communication during times of crisis.

## 3. Engagement and the Potential of Social Networks

Within the context of study of the social networks, new associated terms have appeared which provide conceptual support and symbolic meaning to the phenomenon, to explain the volume of audiences that the networks muster around specific profiles. The exacerbated need to obtain information when facing a situation of crisis creates an unmeasurable anxiety for knowledge, and this allows for the creation of engagement with health professionals who have a presence in the social networks [27]. Engagement, as a conceptual explanation of the phenomenon, implies the massive following of specific profiles, and from a psychological perspective, it implies an interactive and co-creative experience as a response to a stimulus (the COVID-19 crisis, in this case), with respect to an object, i.e., a profile that becomes a communication–information referent [28].

One of the most important emerging issues found in the literature about social media is, without a doubt, engagement, as a psychological state of motivation that results in the act of following. Its study brings us closer to the communication paradigm that emerges from the online environment [29]. This paradigm is especially interesting in a scenario that sees an inflection point of a new communication sphere in the exchange of the producer–consumer and medium–audience roles.

The study of engagement has been addressed from various perspectives as a multi-dimensional phenomenon that explains different types of commitments, and has gained importance with the proliferation of interactions through social networks. According to Barger and Labrecque [30], four different levels of engagement can be established in social networks, having in mind the degree of commitment of the user: (1) observer—content is consumed but there is no interaction or intention to follow; (2) follower—content is consumed and the following of the profile who created it starts, with the first degree of interaction shown as the “follow” action; (3) participant—the user moves to a second degree of interaction in which not only does the user consume the content shared and follow the profile, but also participates actively with likes, retweets/reposts and comments; and (4) defender—this is a greater level of engagement in which aside from following and interacting, content is created and shared that is in favor of the profile or brand that is being followed. It is within the conjunction of the last two types of engagement that the present research addresses the study of audiences in social networks and the establishment of metrics, with indicators depending on the characteristics of each social network analyzed.

## 4. Method

### 4.1. Research Objectives

The phenomenon of COVID-19 as a pandemic has transformed the manner in which the population regards information in general and health information in particular. Starting with the basic premise that the health crisis has triggered the search for information and generalized massive consumption, the objective of this research study is to better understand the relationship established between populations in general and digital media in particular, which has been greatly re-enforced thanks to social networks (Figure 1). Starting with the knowledge that social networks are global in nature, which makes possible the opening of massive communication flows to new actors, it is implied that they can obtain the same or greater audiences, and have a greater convening power and communicative impact as compared to the traditional communication media. Through a comparison between digital communication media and personal accounts of health professionals for a period of six months, a descriptive study is presented of three social networks—YouTube, Twitter and Instagram—utilizing virtual ethnography as the research methodology to verify the hypothesis of the study. For this, the objectives were:To analyze the degree of impact of digital communication media specialized in health as compared to profiles specialized in health information and communication, in terms of passive engagement.To explore cognitive or participative engagement of the flow of interaction created in the networks of digital communication media specialized in health as compared to profiles specialized in health information and communication.

For achieving the objectives, and to obtain a contextualized perspective, an exploratory study is presented in which a methodology based on virtual ethnography was chosen as the research technique, which utilizes social network analysis [31,32]. The presence and evolution of the profiles related with health communication in the forums created by social networks are a communication phenomenon that deserves attention beyond the communication sphere, due to its scope among the public. Therefore, the use of virtual ethnography as the methodology was chosen, as it guarantees an immersion into the area of interaction and socialization of the profiles selected with their audience [33,34].

### 4.2. Selection of the Sample and Methodology

The sample selection was performed having in mind the objectives of the research study. Thus, two different sets of criteria were established due to the different types of samples required for the comparative analysis. The media specialized in health news were selected with the following criteria: (a) they were prominent due to their rigor and degree of specialization, (b) they were well-established in the area of health communication, and (c) they had an important presence in social networks. As for the professional profiles specialized in health information and communication, these were selected with different and valid criteria for performing the comparative analysis. These criteria were: (a) non-institutional profiles, (b) profiles present in social networks, and (c) reliable sources, and with a recognized trajectory in the area of health (with higher education related to health: doctors, pharmacists, and nurses).

For selecting the first set of the sample, the Infonómetro ranking was utilized, as it is created by Infoperiodistas and Acceso (a platform of media, news, agencies, and news professionals), the Federation of Journalist Associations of Spain (FAPE), and the National Association of Health Informers (ANIS), with a total of five media outlets in the top five spots chosen. For the selection of the second set of the sample, the non-probabilistic snowball sampling technique adapted to the study of virtual ethnography was utilized, as it is the most convenient technique for the systematic tracing performed through all the social networks studied. As the technique could negatively affect the study due to bias related to representativeness, the previously defined selection criteria were utilized to avoid this bias. As for community bias, an important secondary bias that could be observed when this type of methodology is utilized, in this case it favored the study, as it demanded similar profiles that met the criteria defined.

The resulting sample, Table 1, was analyzed during a six-month period, organized by trimester. The second trimester encompassed the pandemic crisis and the other encompassed the period prior to it: phase 1 (1 November 2019 to 31 January 2020), first trimester, and, phase 2 (1 February 2020 to 1 May 2020) crisis trimester, starting with the indicators defined for the study of engagement. The data extraction was performed with the tool Social Blade. Also, the data were compared and contrasted in the social networks analyzed for each profile: YouTube, Twitter, and Instagram. The indicators designed for the comparative study were defined starting with previous studies in the field of engagement [27,35], itself defined as the impact or reach of the profile, as well as its ability to attract followers and create interaction. Engagement should be studied at two levels: (1) engagement of passive participation (level two as described by Barger and Labrecque [30]), and (2) cognitive and participative engagement, where interaction is established, for the higher levels of commitment described by the authors: participant and defender.

For the study of both levels of engagement, several types of data were collected with different measurement variables. Also, in the case of level two, specific variables were designed for each social network, taking into account the particularities of their metrics, as shown in Table 2.

It is worth noting that for data collection, the researchers created tracking grids in which they specifically collected data from all variables included in Table 2. For each profile, three different tracking grids were created, one for each social network. For calculating the average increases (index of growth), the percentage of growth was found with respect to the numerical increase of followers or views gained during each trimester. ∑$=\left(\left(V_{1}-V_{2}\right)\times 100\backslash V_{1}\right)$. The starting (V_1_) is the maximum value reached at the end of the trimester and the second value (V_2_) was the minimum value when starting the trimester.

## 5. Study Results

### 5.1. Comparative Analysis of Participation: Passive Engagement

Each social network offers the users possibilities of interaction which allow us to measure the degree of engagement with the profile studied. In this first phase of analysis, the passive participation was studied, which indicates the changes in the number of followers, denoting a superficial level of participation that some authors consider as a low cognitive and participation level [36]. Nevertheless, it provides us with the first comparative description to verify if the specialized communication media, because of their nature, have a greater engagement than the private profiles.

In first place and as expected, the analysis shows that in the case of YouTube (the second-most utilized social network after Facebook), practically all the profiles studied experienced an increase in the number of followers during the crisis trimester as compared to the previous trimesters (Table 3). It is interesting to note that in both groups we found profiles with minimal increases or even negative ones. In the case of ‘Mejor con salud’ (specialized communication media) the increase was 0.16% with respect to the previous trimester, and in the case of ‘Julio Basulto’, the growth was less. In the November to January trimester it grew by 8.28%, while in the crisis trimester it was practically half, only 4.27%.

However, what is more prominent in the case of YouTube, is the degree of growth in the number of followers of ‘Redaccion Medica’, with an increase of 66.83%, indicating that it quadrupled its number of followers since November 2019. ‘Diario medico’ obtained a decent increase of 22.26%, and ‘Con salud’ experienced the greatest increase among the specialized communication media, with an increase of 75.10%, increasing from 1880 followers in November to 10,200 in May. Significant results were also found in the individual profiles chosen, which showed exceptional growth in the case of ‘Boticaria Garcia’, with an increase of 94.13% with respect to the previous trimester, or ‘Spiriman’ with an almost 79% increase, for a total number of followers of 331,000.

Each social network has its own logic, not only for the distribution of content and format, but for the interaction with the users as well. If the creation of content in YouTube is based on videos, a type of mass consumption content whose consumption has increased the most [37], in Twitter the interaction is based on microblogging (posts limited in the number of words—240 characters per tweet—, in which multimodal content is shared: words, images, videos, and links). The engagement with the profiles in this network (Table 4) shows a general increase for all of them, but with a tendency for growth that is very different from the specialized communication media, with an average percentage of 2.45 as compared to the individual profiles, which obtained an average 16.01%. If the specific data is analyzed, it is observed that ‘Boticaria Garcia’ obtained an increase of 31.41%, leading this group along with ‘Spiriman’ (20.21%), and ‘Enfermera saturada’ (23.43%).

Instagram, of the three social networks analyzed, experienced the greatest growth in the last few years, with a greater increase in the number of active accounts. It is characterized by drawing in a younger audience than the others, with 71% of the ‘instagramers’ being under 35 years old [37]. Therefore, the interest in this network comes from the recruitment of a young audience, a differential feature that turns it a network of interest for the traditional communications media, which have serious difficulties in younger audience penetration due to their lack of habits in the consumption of traditional media [38]. In this sense, it is interesting to observe (Table 5) how two members of the specialized media, ‘Diario medico’ and ‘Info salus’, had an extremely modest increase in their number of followers, from 20 to 116 and from 89 to 370 followers, respectively.

The data reveal, once again, a growth in all the profiles studied, with increases between 20 and 55%. However, ‘Boticaria Garcia’, with a growth of almost 30%, gained 100,000 followers between November 1st and May 1st. ‘Julio Basulto’ doubled the number of followers with a figure of 47.46% and ‘Spiriman’ was placed at the head again with increases of 55.57%, multiplying his number of followers by 5, shifting from 102,300 to 528,000, with an audience that is comparable to any broadcasted television program.

Figure 3 shows the growth indices of the sample studied in the three social networks. The growth in the number of followers during the trimester that saw the triggering of the COVID-19 pandemic was general, in some networks more than in others, as shown. Twitter experienced the least growth, especially in the specialized communication media, and the individual profiles loosely grew in all the networks during this period, in some cases with very prominent growth: ‘Con salud’, ‘Boticaria Garcia’ and ‘Spiriman’ with values ≥0.75.

### 5.2. Comparative Analysis of the Flow of Interaction in Networks: Cognitive Engagement

To discover if there was a greater commitment in a participative sense, the analysis of engagement also requires the study of interaction. This is another step in the public-audience-follower relationship that implies being treated as an equal, a “prosumer”, by establishing a dialogue that is unusual in traditional media but has been gaining importance with social audiences [39].

YouTube allows for the analysis of descriptive social metrics, starting with the systematic gathering of the data on the number of views per month offered by Social Blade about each profile. The data (Figure 4) show an increase in the number of views for all the profiles, as expected in this situation; however, the impact of this increase is different in each case. ‘Redaccion medica’ shows an increase of 68.34% as compared with the previous trimester. If the channel had a total of 862,464 views on 1 November 2019, this number almost tripled on 1 May 2020, up to 3,007,196. Another significant increase was found in the profile ‘Con salud’, with a growth of 49.07%, almost doubling the number of views. In the case of the individual profiles, three of them experienced a prominent increase, two of which were extraordinary. In the case of ‘Boticaria Garcia’, the percent of increase was 164.97%, practically quadrupling its number of views, going from 77,106 in November, to 250,476 in May. Another surprising increase was the one experienced by ‘Spiriman’, showing a very significant increase of 248%, reaching views of almost 52 million.

In the case of the individual private profiles, the profile of ‘Lucia mi pediatra’ also experienced a large increase of 24.26%, which is a large increase in views, but somewhat more modest than the cases of ‘Boticaria Garcia’ and ‘Spiriman’. On the other hand, the case of ‘Mejor con salud’ is notable. This channel is well positioned with its public (with a total of 240 million views); however, it should be mentioned that this channel did not react to the pandemic crisis, as in the last ten months, no new content was uploaded to the network, thus its position is the same as prior to the crisis.

The estimation of engagement of any profile in the networks is not complete until the variables found in Table 6 are taken into consideration. After quantifying a low or passive level of engagement, the data showed us variables that reveal the degree of interaction between the profiles studied and their followers. In their approach to the study of engagement, many authors [35,36] point out that the average of ‘likes’ refers to the attitude of the followers about what is shared. Thus, it can be interpreted that a greater number of ‘likes’ signifies a greater affinity and support from the followers. In this sense, it is observed that in the case of Twitter and the profiles of the specialized communication media, they obtained little support from their followers, with values of <5. The situation was very different in the case of the professional health worker profiles, who obtained averages of ≥36 in the case of ‘Boticaria Garcia’ and ‘Julio Basulto’, 134 in the case of ‘Lucia mi pediatra’, and ≥236 in the case of ‘Spiriman’ and ‘Enfermera saturada’. This trend is confirmed if we focus our attention on the ‘average of retweets’ data, which would be within level four ‘defender’ [30]. This level indicates the greatest engagement, within which, aside from following and interacting, the contents shared by the profile followed are created and shared. At the cognitive level, the follower establishes a relationship of commitment in which not only is content consumed, but also supported and disseminated within one’s own network.

The data on Instagram provides us with two variables that can be utilized to analyze the degree of interaction with the profiles, ‘average likes’ and ‘average comments’. Although it is true that in the case of this network, ‘Redaccion medica’, as well as ‘Mejor con salud’ have acceptable averages of likes, 247.20 and 720.24, respectively, the response to the individual profiles is repeated in an overwhelmingly re-enforced manner, also observed in the average of comments, thereby indicating a truly active participation in the case of the individual profiles. Although the degree of engagement does not have to be the same in all the networks, this is how it is in the profiles of the specialized media. The greatest engagement with their followers was found in their YouTube account, i.e., the number of views and the number of comments. However, the degree of engagement does have to be the same in the case of individual profiles. Except for the case of ‘Lucia mi pediatra’, which had a lower average of comments within its category (χ= 1.40), although it was still greater than ‘Diario medico’ (χ = 0.03) and ‘Con salud’ (χ = 1.32), the individual profiles strengthen the trend of the other networks studied, with values of χ = 4.60 in the case of ‘Boticaria Garcia’, 6.30 for ‘Julio Basulto’, χ = 19 ‘Enfermera saturada’, and χ = 1477.90 in the case of ‘Spiriman’, which has a record audience.

## 6. Conclusions

The research presented finds its meaning starting with the context generated on the web after the triggering of the health crisis in Spain in March, 2020. This crisis, on the one hand, alarmed the world’s population, and on the other, confined millions of people in their homes. Both events rocketed the consumption of traditional media as well as internet data and hours of connection [20,22]. In this context, the increase in consumption of social networks was very significant. In Spain, it was calculated that consumption increased by 55%, and a recent study by Comscore, Coronavirus pandemic and online behavioral shifts, revealed that the Spanish population were the most social Europeans, with an increase of 48% in the use of social network apps. The measurement of the phenomenon that took place, in a communications sense, required multiple and very numerous efforts that provided us with different evidence of an unparalleled and unprecedented event. Without a doubt, one of the limitations of the study was the size of the sample, a recurring feature of research studies conducted with virtual ethnography for measuring a scientifically-slippery term such as engagement [12,16]. Nevertheless, the study presented here is a significant contribution to the area of communication, due to its specific empirical evidence, a snapshot of how the hegemony of the traditional media in the crisis information discourse was profoundly transformed by the phenomenon of social networks, and how this transformation was accelerated due to a health crisis [9,10,11]. The concept of prosumer is established from a social audience [39], leaving behind the passivity of the masses who expectantly consume the media content. The new audiences need interaction and need to be part of the media flow, as pointed out by Helbing [19], assuming that the attraction of the online environments makes us all social and participative beings. Not only this, but the change in roles is carried out by individual profiles that are able to muster audiences that are comparable to the share garnered by popular programs, such as in the case of ‘Spiriman’ on YouTube.

One of the main conclusions re-enforced by the study presented is that the transformation of the media flow towards interactive and global activity, open to all actors, indicates that traditional communication is no longer unique [16], and this has been highlighted more than ever with the health crisis brought on by COVID-19. The situation has been personified by the main role of the social networks as alternative or parallel spheres for finding information and the consumption of content, as previously discussed in other works [40]. The data from this study re-enforce this idea, making us conclude that, despite the specialized communication media studied being active in social networks, with an average number of 121,252 tweets, the attitude of the followers showed an average of likes that was 149 points below that obtained by the health professional profiles with half the number of tweets published. The data from YouTube also support this finding and trend, although it is more prominent: as compared to the number of videos uploaded by communication media specialized in health, 20,483, with an average of four comments, the health professional profiles, despite the videos published being 72 times less (281), garnered an average of 300 comments.

Aside from the great challenge this crisis has brought from the point of view of fake news and hoaxes, defined by some social actors as the greatest challenge faced by fact checkers, the recent research studies have shown the need to work in this direction [10,15,18]. In this sense, the new professional opportunities are open to a new communications paradigm. The COVID-19 crisis has accelerated, without a doubt, a change that has already been described in terms of new literacies and new communication paradigms [3,41], providing conclusions that imply various stimulating challenges in the communications paradigm. On the one hand, the essential functions of the communication media are broadened, which consolidate their position as gatekeepers of the truth; this is translated as new roles for the journalist, and new horizons of growth are opened for the communications industry. On the other hand, the yearned-for digital transformation as a project for the future before the crisis, has become established in an unusual manner as a strategy for survival, as the networks legitimize new actors on the communications stage. The need to work online has compelled communications professionals towards a naturalized plethora of videoconferences, interventions from home, digital montages, interviews from different spaces—interventions that put everyday realities into context. This has naturalized the ‘youtuber’ aesthetic, pushing society even more into a media convergence that includes the social networks as well-positioned actors of the communication flow, as previously described in prior works [16,28,42]. The pandemic has accelerated events, evidencing a communications model that emerges with an indisputable collaborator: the social networks, dependent on the IoT (Internet of Things), based on the advanced analysis of big data with powerful allies such as deep learning and artificial intelligence. A new paradigm is becoming established and the health crisis has contributed to emphasizing and underlining this phenomenon, which must be understood and adopted as a challenge for the communications industry, an opportunity for professionals, and a need for the renovation of learning from the educational system itself [43].

## Figures and Tables

**Figure 1 ijerph-17-05261-f001:**
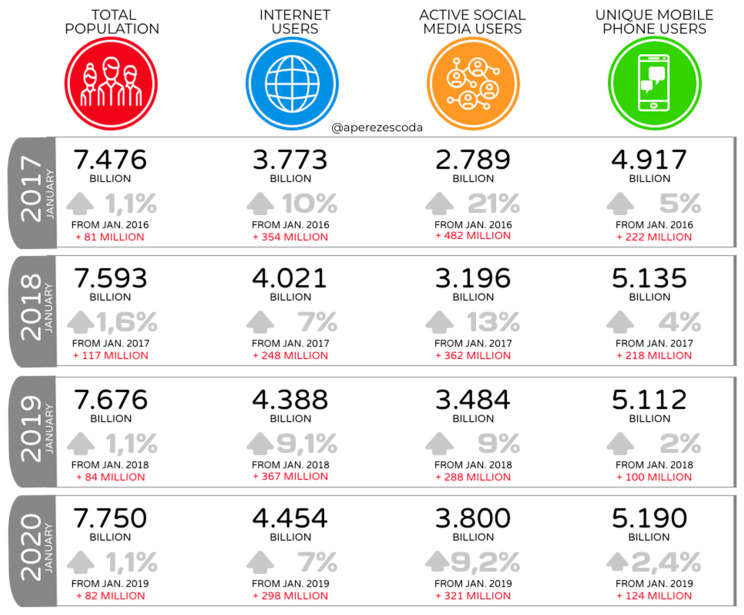
Annual change worldwide of the world’s population, internet users, social network users, and mobile phone users. Created by the author from the annual reports from Hootsuite (We are social, 2017, 2018, 2019, 2020).

**Figure 2 ijerph-17-05261-f002:**
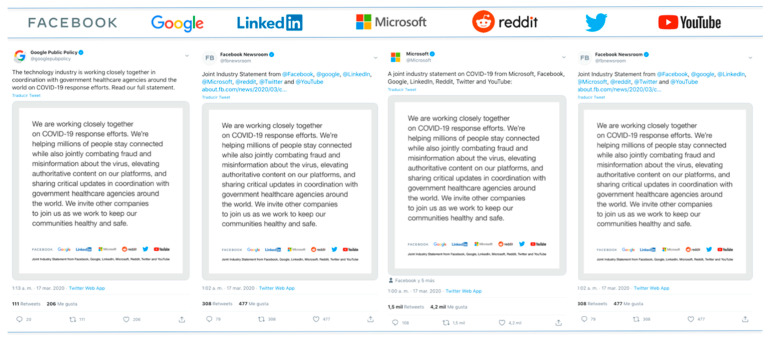
Joint declaration of the technology industry in an effort to provide a response to the COVID-19 pandemic (Facebook Newsroom, 2020). Created by the author from the tweets published by Google, Facebook, Microsoft, and Reddit.

**Figure 3 ijerph-17-05261-f003:**
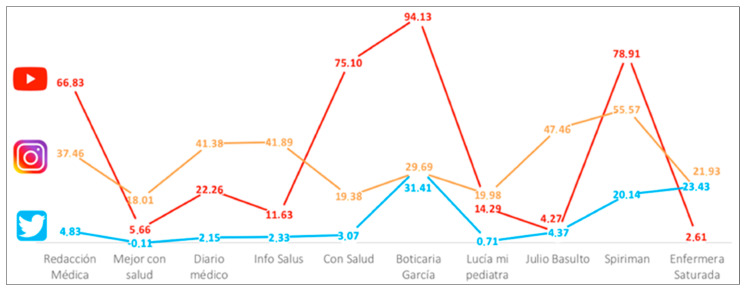
Summary of the growth indices in the number of followers in the February 2020–May 2020 trimester (pandemic crisis) with respect to the previous trimester, November 2019–January 2020.

**Figure 4 ijerph-17-05261-f004:**
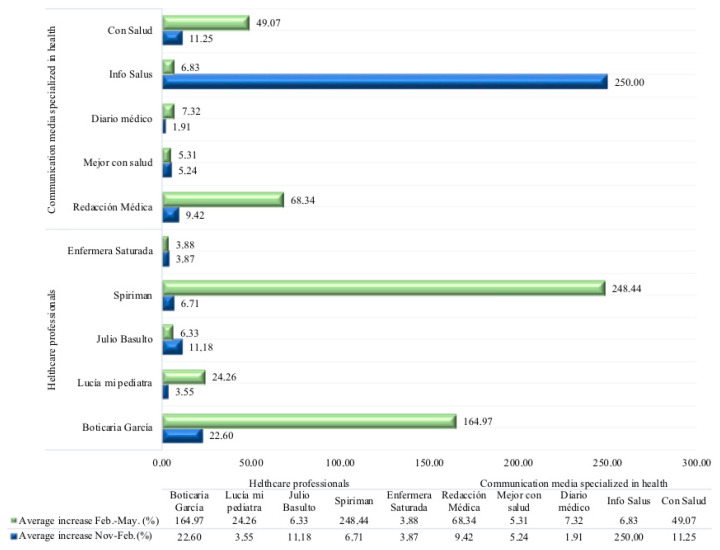
Chronological change of the growth index of interactions in the social network YouTube in the two trimesters analyzed.

**Table 1 ijerph-17-05261-t001:** Description of the sample object of study comprised of five communication media specialized in health information and five healthcare professional profiles.

Profiles	Name	Name in Social Networks	Twitter	YouTube	Facebook	Instagram	Webpage
Communication media specialized in health	Redacción médica	@redaccionmedica	72.9 mil	8.11 mil	66,538	11.9 mil	https://www.redaccionmedica.com
	Mejor con salud	@mejorconsalud	59.1 mil	106 mil	9,608,458	65.2 mil	https://mejorconsalud.com
	Diario médico	@diariomedico	116 mil	1.06 mil	26,029	116	https://www.diariomedico.com
	Infosalus	@infosalus_com	56.6 mil	344 mil	34,084	370	https://www.infosalus.com
	Consalud	@Consalud_es	120.6 mil	10.2 mil	224,700	8102	https://www.consalud.es
Healthcare professionals	Marian García	@boticariagarcia	53.4 mil	6850	54,324	358 mil	https://boticariagarcia.com
	Lucía mi pediatra	@luciapediatra	66.7 mil	4840	3045,024	294 mil	https://www.luciamipediatra.com
	Julio Basulto	@JulioBasulto_DN	77.5 mil	17,000	108,585	192 mil	https://juliobasulto.com
	Jesús Candel	@Spiriman	125.5 mil	331,000	33,809	528 mil	https://www.justiciaporlasanidad.org
	Héctor Castiñeira	@EnfermeraSaturada	134.5 mil	3070	326,527	226 mil	http://www.enfermerasaturada.es/

**Table 2 ijerph-17-05261-t002:** Study variables.

Study Variables			
Level 1. Passive participation Engagement	YouTube	Number of subscribers or followers	Followers 1 November 2019
			Followers 1 February 2020
			Followers 1 May 2020
		Index of growth first trimester	
		Index of growth during the crisis trimester	
	Twitter	Number of subscribers or followers	
		Index of growth first trimester	
		Index of growth during the crisis trimester	
	Instagram	Number of subscribers or followers	
		Index of growth first trimester	
		Index of growth during the crisis trimester	
Level 2. Cognitive participative Engagement	YouTube	Number de views *	Views 1 November 2019
			Views 1 February 2020
			Views 1 May 2020
		Index of growth first trimester *	
		Index of growth during the crisis trimester *	
		Number of comments in the last 50 videos **	
		Mean of comments **	
		Number of publications	
	Twitter	Number of followers *	
		Number likes *	
		Mean of likes *	
		Number of tweets	
		Mean of retweets **	
	Instagram	Number of followers	
		Degree of engagement	
		Mean of likes *	
		Number of publications	
		Mean of comments **	

* Variables describing participant cognitive participative engagement ** Variables describing defender cognitive participative engagement.

**Table 3 ijerph-17-05261-t003:** Chronological change of the number of followers in the social network YouTube.

Names and Profiles		Followers 1 November 2019	Followers 1 February 2020	Followers 1 May 2020	Average Increase November–February (%)	Average Increase February–May (%)
Communication media specialized in health	Redacción Médica	2410	2690	8110	10.41	66.83
	Mejor con salud	94,500	100,000	106,000	5.50	5.66
	Diario médico	756	824	1060	8.25	22.26
	Info Salus	277,000	304,000	344,000	8.88	11.63
	Con Salud	1880	2540	10,200	25.98	75.10
Healthcare professionals	Boticaria García	353	372	6340	5.11	94.13
	Lucía mi pediatra	3830	4140	4830	7.49	14.29
	Julio Basulto	14,400	15,700	16,400	8.28	4.27
	Spiriman	68,000	69,800	331,000	2.58	78.91
	Enfermera Saturada	2940	2990	3070	1.67	2.61

**Table 4 ijerph-17-05261-t004:** Chronological change of the number of followers in the social network Twitter.

Names and Profiles		Followers 1 November 2019	Followers 1 February 2020	Followers 1 May 2020	Average Increase November–February (%)	Average Increase February–May (%)
Communication media specialized in health	Redacción Médica	68,520	69,454	72,981	1.34	4.83
	Mejor con salud	59,026	59,228	59,162	0.34	−0.11
	Diario médico	111,475	113,572	116,072	1.85	2.15
	Info Salus	54,869	55,375	56,696	0.91	2.33
	Con Salud	114,963	116,965	120,667	1.71	3.07
Healthcare professionals	Boticaria García	45,090	36,095	52,621	−24.92	31.41
	Lucía mi pediatra	66,388	66,233	66,706	−0.23	0.71
	Julio Basulto	70,867	74,212	77,600	4.51	4.37
	Spiriman	118,473	100,243	125,524	−18.19	20.14
	Enfermera Saturada	99,400	103,123	134,676	3.61	23.43

**Table 5 ijerph-17-05261-t005:** Chronological change of the number of followers in the social network Instagram.

Names and Profiles		Followers 1 November 2019	Followers 1 February 2020	Followers 1 May 2020	Average Increase November–February (%)	Average Increase February–May (%)
Communication media specialized in health	Redacción Médica	5420	7504	11,998	27.77	37.46
	Mejor con salud	44,876	53,525	65,282	16.16	18.01
	Diario médico	50	68	116	26.47	41.38
	Info Salus	89	215	370	58.60	41.89
	Con Salud	5880	6532	8102	9.98	19.38
Healthcare professionals	Boticaria García	159,018	182,582	259,674	12.91	29.69
	Lucía mi pediatra	201,426	236,423	295,452	14.80	19.98
	Julio Basulto	88,740	101,158	192,522	12.28	47.46
	Spiriman	102,300	234,600	528,000	56.39	55.57
	Enfermera Saturada	171,770	176,862	226,533	2.88	21.93

**Table 6 ijerph-17-05261-t006:** Indicators of Engagement in the three social networks studied.

Names and Profiles		Twitter Interaction					Instagram Interaction				YouYube Interaction			
		Followers Number	Likes Number	Likes Average	Tweets Number	Retweet Average	Engagement	Likes Average	Publications Number	Comments Average	Views Number	Comments in Last 50 Videos	Average	Publications
Communication media specialized in health	Redacción Médica	72,981	7096	2	237,189	2	2.09	247.20	666	3.92	3,092,491	529	10.60	4165
	Mejor con salud	59,162	23	1	40,325	4	1.15	740.24	3594	9.64	240,143,294	380	7.60	592
	Diario médico	116,072	1566	4	41,299	5	0.00	11.34	5	3.00	940,707	16	0.03	300
	Info Salus	56,696	21,219	4	62,308	5	0.00	20.10	0	0.20	408,696,478	178	3.56	97,148
	Con Salud	120,667	71,867	2	225,141	2	0.67	53.48	618	0.44	293,336	66	1.32	211
Healthcare professionals	Boticaria García	52,621	43,861	36	69,528	8	2.37	5869.24	2106	285.56	250,476	230	4.60	71
	Lucía mi pediatra	66,706	79,389	134	27,806	51	1.39	3964.84	3291	150.80	188,124	71	1.40	36
	Julio Basulto	77,600	138,450	43	140,420	18	1.17	1261.28	7612	59.20	1,136,228	315	6.30	277
	Spiriman	125,524	11,922	230	9192	152	8.49	5340.80	1194	420.67	51,909,906	73.894	1477.90	More 1000
	Enfermera Saturada	134,676	28,215	318	22,264	169	5.12	11249.60	1239	350.12	342,259	363	1900	19

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
