# Peer review of "Social Networks’ Engagement During the COVID-19 Pandemic in Spain: Health Media vs. Healthcare Professionals"

_ijerph, 2020, doi:10.3390/ijerph17145261_

Round 1

Reviewer 1 Report

I read the paper "Social Networks Engagement During the Covid-19 Pandemic in Spain: Health Media vs. Healthcare Professionals" with great interest. The topic is very timely. The work presents an analysis of the phenomena that have occurred in the last six months and concerns the impact of a pandemic on social networks engagement, and a comparison of this involvement in the case of various profiles belonging to Health media and those run by individual medical professionalists. The publication is written very well, it comprehensively presents the analyzed problems. Certainly deserves to be published.

Nevertheless, I have to make some critical remarks and ask the authors to respond to them before accepting the paper for publication.

Major remarks:

1/ The authors comprehensively present the differences between selected profiles in the context of differences in the engagement of Internet recipients through three social media - YouTube, Instagram and Twitter.

However, I lacked an analysis of this (explanatory hypotheses) as to why these differences may arise. Of course, this was not the main subject of the study, but it is lacking as a complement to the topic - perhaps as an indication to which direction should subsequent studies / analyzes go. What can be the reason for the observed differences between Health Media and Healthcare Professionals? After the pandemic, do the authors expect these differences to persist? Could this be influenced by the content posted on the profiles? Or other factors could have contributed to this? Some possible factors affecting this differentiation are mentioned when describing the results of the research - it would be good if they were also included in the discussion analyzing the results and summarizing the work.

Minor remarks:

1/ line 81-

what is the difference between user and active user – please describe here

2/ line 100 and 101-

 the number of giants has decreased from 7 to 5, and therefore the market for service providers has become more concentrated-  (concentration increased and not decreased) – maybe my doubt comes from a poor sense of language differences?

3/paragraphs beginning in verses 164 and 167-
I'd precede them with bullets to make them more visible

4/ In the section Selection of the sample and methodology on page 219 - in verses 203 and 204,

 the authors indicate the study periods - I would add the term - Phase 1 (dates before the epidemic) - first trimester and phase 2 (dates after the epidemic) - crisis trimester. In the text, the authors first mention the crisis, then the period before it, and then give it in the reverse order, which is misleading then when they write about the first phase in the rest of the manuscript.
This would organize later reference to these periods

5/ in verse 213-

 instead of the term "different" I would prefer "several" rather

6/ Table 2-

 it is worth considering removing "number of followers" from Level 2, as the characteristics describing cognitive participative engagement should be placed here. In addition, it may be worth specifying here what is associated with the term 3 / participant and the term 4 / defender.

7/ Table 3

- it seems to me that there should be a percentage increase and not an "average increase" - I am not sure what to calculate the average for here.

8/ In verse 237

I would rather suggest / YouTube network” and not “This network”

9/ Figure 3 –

not everyone knows the logo of individual social media - this text has not been described anywhere else in this text and it has not been added here in the legend - it should be added so that the reader can identify these media in the figure.

My comments do not diminish the value of the manuscript. In some passages it is only difficult to read - maybe it results from the necessity of language editing.

Reviewer 2 Report

As the authors claim, the COVID-19 pandemic has significantly altered social media behaviors. The topic of this manuscript is significant and interesting, especially for health information delivery and broadcasting. This study chose nine social media accounts with two types 1) communication media specialized in health, 2) healthcare professionals. Based on the collecting data, including followers, several likes, retweets numbers, etc, form Twitter, Instagram, and YouTube, authors compared the differences between the two types of accounts. Detailed comments are listed as follows:

  1. Why chose these nine accounts representing the Health Media and Healthcare Professionals? Please explain this in more detail.
  2. How and when did the social media usage data gather?
  3. Health Media and Healthcare professionals are entirely two distinct types of accounts. First, Health Media generally are belonged to news agencies or organizations. The interactions on social media may indicate the altitude or gesture of agencies or organizations. Hence, real persons using these Health Media accounts are unable to often interact with the audiences. But the Healthcare Professionals are the opposite. Is the comparison of this manuscript appropriate?

Second, social media users or audiences have various expects from this Health Media and Healthcare Professionals. Health Media is more like a heal information provider or supplier for general users who are stratified by being informed. In contrast, individuals prefer detailed consultation and advice from Healthcare Professionals. These differences result in the different interaction patterns between social media users and the two types of accounts.

Therefore, my primary concern is it may not be suitable to directly compare these two types because they have different backgrounds and roles.

  1. A lot of typos and formatting errors exist in the manuscript. For example:
  • ‘University of Nebrija’ repeated twice in each affiliation address
  • Missing spaces in the manuscript (e.g., Line 71)
  • Confusing use of full stop in the number of Tables.
  • Keep the consistency of words: ‘Con Slaud’,’Con slaud’
  • ‘Tuits’ should be ‘Tweets’ (Table 6)
  • Please check the length of the sentences. Too long and complex sentences have poor reading experience.

Reviewer 3 Report

Thank you for your well-written and timely manuscript about COVID 19 and social media communication.

My overall comment is that it takes 5 to 6 pages before you really introduce the research question and what you intend to do. While the background about social media is interesting and useful, I think you could reduce these sections and add a paragraph or two in the introduction to explain the purpose of this manuscript - research question, brief overview of the methods, and organization of the paper.

There were some minor grammatical errors that could be cleaned as well.

Round 2

Reviewer 2 Report

The authors have answered my concerns. This manuscript can be published in IJERPH.